# Dairy Consumption at Snack Meal Occasions and the Overall Quality of Diet during Childhood. Prospective and Cross-Sectional Analyses from the IDEFICS/I.Family Cohort

**DOI:** 10.3390/nu12030642

**Published:** 2020-02-28

**Authors:** Iris Iglesia, Timm Intemann, Pilar De Miguel-Etayo, Valeria Pala, Antje Hebestreit, Maike Wolters, Paola Russo, Toomas Veidebaum, Stalo Papoutsou, Peter Nagy, Gabriele Eiben, Patrizia Rise, Stefaan De Henauw, Luis A Moreno

**Affiliations:** 1Growth, Exercise, Nutrition and Development (GENUD) Research Group, University of Zaragoza, 50009 Zaragoza, Spain; iglesia@unizar.es (I.I.); lmoreno@unizar.es (L.A.M.); 2Instituto Agroalimentario de Aragón (IA2), 50013, Zaragoza, Spain; 3Instituto de Investigación Sanitaria Aragón (IIS Aragón); 50009 Zaragoza, Spain; 4Red de Salud Materno-infantil y del Desarrollo (SAMID), Instituto de Salud Carlos III, Madrid, 50009 Zaragoza, Spain; 5Leibniz Institute for Prevention Research and Epidemiology—BIPS, 28359 Bremen, Germany; intemann@bips.uni-bremen.de (T.I.); hebestr@leibniz-bips.de (A.H.); wolters@bips.uni-bremen.de (M.W.); 6Institute of Statistics, Bremen University, 28359 Bremen, Germany; 7Centro de Investigación Biomédica en Red de Fisiopatología de la Obesidad y Nutrición (CIBERObn), Instituto de Salud Carlos III, 28029 Madrid, Spain; 8Epidemiology and Prevention Unit, Department of Preventive & Predictive Medicine, Fondazione IRCCS Istituto Nazionale dei Tumori, 20133 Milan, Italy; valeria.pala@istitutotumori.mi.it; 9Epidemology & Population Genetics, Institute of Food Sciences, National Research Council, 83100 Avellino, Italy; prusso@isa.cnr.it; 10Department of Chronic Diseases, National Institute for Health Development, 11619 Tallinn, Estonia; toomas.veidebaum@tai.ee; 11Research and Education Institute of Child Health, 2018 Strovolos, Cyprus; stalo.papoutsou@gmail.com; 12Department of Pediatrics, University of Pécs, 7622 Pécs, Hungary; peter.nagy@kk.pte.hu; 13Department of Public Health and Community Medicine, University of Gothenburg, 405 30 Gothenburg, Sweden; gabriele.eiben@medfak.gu.se; 14Department of Biomedicine and Public Health, School of Health and Education, University of Skövde, 541 28 Skövde, Sweden; 15Department of Pharmacological Sciences, University of Milan, 20122 Milan, Italy; patrizia.rise@unimi.it; 16Department of Public Health, Ghent University, 9000 Ghent, Belgium; Stefaan.DeHenauw@UGent.be

**Keywords:** dairy, snack, diet quality, children

## Abstract

There is scarce information on the influence of dairy consumption between main meals on the overall diet quality through childhood, constituting the main aim of this research. From the Identification and prevention of Dietary- and lifestyle induced health EFfects In Children and infantS (IDEFICS) study, and based on the data availability in each period due to drop outs, 8807 children aged 2 to 9.9 years from eight European countries at baseline (T0: 2007–2008); 5085 children after two years (T1); and 1991 after four years (T3), were included in these analyses. Dietary intake and the Diet Quality Index (DQI) were assessed by two 24 hours dietary recalls (24-HDR) and food frequency questionnaire. Consumption of milk and yogurt (*p* = 0.04) and cheese (*p* < 0.001) at snack meal occasions was associated with higher DQI scores in T0; milk and yogurt (*p* < 0.001), and cheese (*p* < 0.001) in T1; and cheese (*p* = 0.05) in T3. Consumers of milk (*p* = 0.02), yogurt (*p* < 0.001), or cheese (*p* < 0.001) throughout T0 and T1 at all snack moments had significantly higher scores of DQI compared to non-consumers. This was also observed with the consumption of cheese between T1 and T3 (*p* = 0.03). Consumption of dairy products at snack moments through childhood is associated with a better overall diet quality, being a good strategy to improve it in this period.

## 1. Introduction

Dietary factors in relation with obesity at different life stages from early childhood to adolescence are barely understood [1]. Recently, some eating patterns such as eating frequency, snacking, or skipping meals have been suggested to influence energy intake regulation [2].

In fact, for this specific age target, the consumption of at least four meal occasions per day is recommended, whereas the effect of eating five or more meals per day remains to be elucidated [3]. However, composition of snack meal occasions and their association with total dietary intake or body composition have not been investigated in depth during childhood [4].

The main dairy products consumed are (1) yogurt, which is defined as a food in the form of a thick, slightly sour liquid that is made by adding bacteria to milk and (2) cheese. Other dairy products consumed were included in milky desserts, which are a pooled group of foods that contain a lot of milk, and sweetened milk, which has an added sugar portion inside. Dairy products are good sources of calcium, which helps maintain bone mineral content and could reduce the risk of fractures later in life [5].

Some studies have also suggested that dairy consumption may have a protective effect against the development of obesity and cardiovascular diseases (CVD) [6]. Yet, the literature in children and adolescents in this sense is scant; in general, studies indicate either a beneficial or a neutral effect of dairy foods or calcium consumption on body weight or body composition [6,7].

Besides, an understanding of the tracking of dietary intake appears to be relevant for formulating policies focused on nutrition related health outcomes such as obesity. In dietary survey, tracking is defined as similar levels of dietary intake over time [8]. Despite childhood and adolescence being critical periods for physical development, not many studies have investigated the tracking of dietary intake during these periods.

In order to facilitate the appraisal of adequate dietary intake, diet quality indices (DQIs) have been used in various studies. DQIs have shown to provide an adequate overview of an individual’s dietary intake and consequently to assess the compliance with food-based dietary guidelines (FBDG). A number of diet quality indicators have arisen, but their associations with health-related outcomes are still debated [9]. For instance, Vyncke et al. [10] showed good validity of the DQI for adolescents when comparing food and nutrient intakes and biomarkers in European adolescents using that index. Nevertheless, up to date, there is no study assessing the association of the consumption of dairy products with these DQIs and likewise none considering specific occasions of consumption as snack meal occasions considering the childhood period so widely.

Therefore, the objective of this study is to describe food consumption focusing on dairy products at snack meal occasions (mid-morning, mid-afternoon, after-dinner) during childhood and to investigate their relationship with the overall diet quality (cross-sectionally and longitudinally).

## 2. Material and Methods

### 2.1. Study Subjects

This study is based on the “Pan-European IDEFICS/I.Family children cohort”, which can be found at the ISRCTN registry under the ID ISRCTN62310987. The “Identification and prevention of Dietary- and lifestyle induced health EFfects In Children and infantS” (IDEFICS) cohort comprised a total of 16,228 children aged between 2 and 9.9 years (y) old at baseline (T0) [11,12,13]. They were examined from autumn 2007 to spring 2008 in a community based baseline survey (T0) in eight European countries and also in a follow-up performed in 2009–2010 (T1) and from 2013 to 2014 in the “Investigating the determinants of food choice, lifestyle and health in European children, adolescents and their parents” (I.Family) (T3) study. Final included samples in the follow-ups comprised 11,041 and 6055 children of the original IDEFICS cohort in T1 and T3, respectively [11,12,13]. The design of the study was not thought to provide a representative sample for each country but constituted the starting point of the largest European children’s prospective cohort study established up to date, involving Sweden, Germany, Hungary, Italy, Cyprus, Spain, Belgium, and Estonia. In each country, two separated cities or regions were involved in the study to have an intervention and a control region. This community intervention was carried out between autumn 2008 and spring 2009.

Children from 2 to 9.9 y who lived in the specific regions and who belonged to the selected pre-schools, kindergartens, or primary schools were eligible for recruitment. Children were approached via schools and kindergartens. In addition to the informed consent signed by parents, each child was asked to give verbal approval immediately before examination (or proper informed consent if they were older than 12 y in T3). In each country, participating centers obtained ethical approval from the local responsible authorities in accordance with the ethical standards laid down in the 1964 Declaration of Helsinki and its later amendments [14].

The general purpose of both studies IDEFICS and I.Family was to understand the mechanisms of childhood obesity and how to prevent it and to identify determinants of eating habits, lifestyles, and health in European children and their families. The attrition rate was defined by Langeheine et al. [14]. Children’s age and weight status was positively associated with attrition; however, mother’s age, migrant background, and educational level were associated with lower attrition. Taking into consideration the availability of the variables for this study [age, sex, BMI z-scores, education of the families, region, valid data on the two dietary assessment methods 24-HDR and food frequency questionnaire (FFQ)], total samples in each of the surveys were 8807 in T0, 5085 in T1, and 1991 children in T3 for cross-sectional analyses. In relation to longitudinal analyses, we finally included: 3838 children for tracking between T0 and T1; 974 children for tracking between T1 and T3; and 977 children with available data to assess the tracking between T0 and T3. In Figure 1, the flow chart of the selection process is provided. 

### 2.2. Dietary Assessment

For the purpose of this study, dietary intake was assessed using the computer-assisted 24-h dietary recall (24-HDR), called SACINA (“Self-Administered Children and Infant Nutrition Assessment”) in T0 and T1 and SACANA (“Self-Administered Children, Adolescents, and Adult Nutrition Assessment”) in T3. SACANA was the revised and extended web-based version of the SACINA off-line 24-HDR. SACINA was based on the previous HELENA-DIAT (“HELENA-DIetary Assessment Tool”) software [15] developed within the HELENA Study (http://www.helenastudy.com) for adolescents. The absolute validity of proxy-reported energy intakes from SACINA [16] was investigated based on comparison with total energy expenditure measured using the doubly labeled water technique. SACANA was validated twice. In IDEFICS, both at T0 and T1, parents or caregivers were those who reported the information on amount and type of all foods and drinks that were consumed by the children during the previous weekday. In T3, this was done by the children themselves with the help of a parent from the age of 8 y. Estimation of portion size was assisted using standardized photographs. For children who had lunch in the canteen, school meals were assessed using a standardized observer sheet, completed by trained staff. Country-specific food composition tables (FCT) were used to match simple foods or recipes [17]. All nutrients and energy values were expressed per 100 g edible portion. Standard units were taken from McCance and Widdowson’s [18] food composition tables (FCT). As the Italian FCT provided nutrient data only for raw foods, a raw/cooked coefficient was applied when large raw/cooked deviations were expected after preparation by boiling or steaming [16].

In a subsample, in T0 and T1, two 24-HDR were recorded, but to not lose participants in this study, only the first 24-HDR was considered in each child (also in T3).

SACINA instrument offered food and beverage reporting during six meal occasions: breakfast, mid-morning snack (everything recalled from breakfast to lunch), lunch, afternoon snack (everything recalled from lunch to dinner), evening meal, and evening snack (everything recalled after dinner) [16]. It was also possible to add more snack occasions.

Missing or implausible values that could not be corrected were imputed by country, food group, and age-specific median intakes (this represented 0.01% of the entries for SACINA and 0.14% for SACANA). Incomplete 24-HDR and those with four or more imputed values were excluded from the analysis. 

For the cross-sectional analyses in all periods, intake (in grams per day) of the food items that belongs to the food group of dairy products consumed at mid-morning-snack, mid-afternoon snack, after dinner, and during the day, and the total energy intake (TEI) in kcal per day were provided. In the category “milk”, whole milk, skimmed, and semi-skimmed milk were included; in “yogurt”, all types of yogurt; in “cheese”, all types of cheese; in “milky desserts” were products such as junket or rice pudding; and in “sweetened milk”, there were products such as milk-shakes. “Milky desserts + sweetened milk” considered together as a category, and cheese, were only described for main meal occasions and considering all snack occasions together due to their low consumption.

The food frequency questionnaire (FFQ), which was part of the Children’s Eating Habits Questionnaire (CEHQ) in the IDEFICS study [19], was also used in the cohort to calculate the Diet Quality Index (DQI) in each survey. The FFQ was found to provide reproducible and valid data [19,20]. In T0 and T1, the CEHQ included 43 food items clustered into 36 according to their nutritional profiles, as it was previously described [19], and also to make them comparable to the food categories of the 24-HDR. Similarly, in T3, the FFQ included the food items from T0 and added 17 food items due to older children and adolescents usually eating a wider range of food groups than the youngest. All food items were also re-arranged with the same nutritional profile purpose in 43 categories.

### 2.3. Diet Quality Index

The Diet Quality Index (DQI) was used in this study to account for the overall diet quality of the children´s diet. This index was originally developed for preschool children [21] and was later adapted and validated for its use in adolescents [10]. It consists of three components: dietary quality, dietary diversity, and dietary equilibrium. The basis to compute it was described previously in Vyncke´s and in Huybrechts et al.’s research [10,21]. This index was previously used as an indicator of compliance with the Flemish Food Based Dietary Guidelines (FBDG), which are similar to the dietary guidelines in other countries and in the World Health Organization CINDI (Countrywide Integrated Non-communicable Disease Intervention program) pyramid [10].

Dietary quality component expressed whether the food consumption of the child was of optimal quality within a food group. It was represented by a “preference group”, an “intermediate group” and a “low-nutrient, energy-dense group”. This classification can be observed in Appendix A
Table A1. To calculate this dietary quality component, the amount of food groups consumed was considered. As we used a qualitative FFQ, sex-, age-, and country-specific medians of the portion sizes of the corresponding food groups were derived based on the 24-HDR. Portion sizes were multiplied by the derived frequencies from the FFQ to get the amount of food groups consumed daily, according to Bel-Serrat et al., Koster-Rasmussen et al., and Marcinkevage et al. [19,22,23].

Dietary diversity expressed the degree of variation in the diet by giving points ranging from 0 to 9 for each different serving of food consumed from the recommended food group (those from the list of preferable groups in Appendix A
Table A1).

Finally, dietary equilibrium was calculated from the difference between the adequacy component (which is the percentage of the minimum recommended intake for each of the main food groups) and the excess component (which is the percentage of intake exceeding the upper level of the recommendation) [10]. The actual recommendations to calculate the adequacy and the excess of each food group are based on the Flemish Food Based Dietary Guidelines (FBDG) [24] for the different age ranges: <10 y and ≥10 y.

These three components of the DQI are presented in percentages. The dietary quality component ranged from −100 to 100%, while dietary diversity and dietary equilibrium ranged from 0 to 100%. To compute the DQI, the mean of these components was calculated. As such, the DQI ranged from 33 to 100%, with higher scores reflecting higher diet compliance. The score was calculated at baseline, T1, and T3 surveys. DQI scores for an individual provide an estimate of diet quality relative to dietary guidelines.

### 2.4. Body Mass Index

All measurements followed standard and validated procedures in T0, T1, and T3 surveys. Children were weighted in fasting status, without shoes and with light clothing on a body composition analyzer (Tanita BC 420 MA specifically adapted for children’s feet in T0 and T1, and TANITA BC 418 MA in T3), recorded to the nearest 0.1 kg. Height was measured with a stadiometer (SECA 225 in T0 and T1, and SECA 213 in T3, Birmingham, UK) and recorded to the nearest 0.1 cm. Body mass index (BMI) was calculated by dividing body weight in kilograms by squared body height in meters and transformed to an age- and sex-specific z–score according to Cole et al. [25].

### 2.5. Sociodemographic Factors

Highest educational level of the parents according to International Standard Classification of Education (ISCED) was used as proxy indicator for socio-economic status of the family [26].

### 2.6. Intervention

Intervention was designed to address key obesity-related behaviors (diet, physical activity, and stress) in 4 levels: individual (children), family (parents), schools, and community. Briefly, six messages related to these behaviors were delivered through 10 modules targeted across all the levels: (1) increase water consumption; (2) increase the consumption of fruits and vegetables; (3) reduce daily screen time; (4) increase daily physical activity; (5) improve the time of family quality; (6) ensure adequate sleep duration. More information in this respect can be found elsewhere [27]. As this intervention took part between T0 and T1, we used a covariate to avoid the possible effect this might have in our analyses.

### 2.7. Statistical Analysis

The distribution of the sample regarding the categories of parents’ education at baseline (T0) and in the two follow-up examinations (T1 and T3) as well as per country and per region (intervention or control) were presented stratified by sex and age group. Similarly, descriptive data of the study sample for the BMI z-score, diet quality, total energy intake, total dairy intake, and the food items eaten at snack meals and main meals occasions were presented as means and standard deviations for each survey and age group and sex.

Multiple linear regression analyses adjusted by BMI z-score, total energy intake, education of parents (the category of the one with highest education level) in the last survey of the tracking (T1 or T3), and region (control vs. intervention) were used to determine the differences in DQI between those who had consumed dairy products at snack occasions and those who did not. Similarly, for the prospective analysis, the dairy consumption at snack occasions was categorized in several groups of “consumers” at T0 and T1: dairy consumers at T0 and T1, non-consumers either in T0 and T1, dairy consumers in T0 but not in T1, and finally, non-dairy consumers in T0 but consumers in T1. After the creation of these groups, multiple linear regression analyses were again applied to investigate the differences in DQI between the healthy (consumers in both surveys and non-consumers in the first survey but consumers in the final survey, apart from the case for milky desserts and sweetened milk, for which it was the opposite) and unhealthy (non-consumers in both surveys, consumers in the first survey but non-consumers in the final survey, apart from the case for milky desserts and sweetened milk, for which it was the opposite) tracking groups after adjustment by the already mentioned variables. The same was done for T0–T3 and T1–T3 tracking surveys. The Statistical Package for Social Sciences version 22.0 (SPSS Inc., Chicago, IL, USA) was used for analyses. All statistical tests were stratified by sex and age group (<6 y and 6–9.9 y for T0 and T1, and <10 y and ≥10 y for T3), and corresponding p values were two-sided, considering *p* < 0.05 as statistically significant.

## 3. Results

Figure 2, Figure 3 and Figure 4 are presented in order to easily observe the mean amounts of main groups of dairy products consumed during all snack occasions per country in T0, T1, and T3, respectively. In general, the main obtained observation is the drop out of the consumption of dairy products at these occasions. In T0 (Figure 2), milk and yogurt are the categories more consumed in snack occasions. However, in Cyprus and Spain, sweetened milks are more consumed than yogurt, while in Hungary and Germany, sweetened milk is the most consumed category of dairy products.

In T1 (Figure 3), milk is the most consumed dairy product at snack occasions for Italy, Estonia, Cyprus, and Sweden, while yogurt is for Belgium and Spain. In Cyprus, sweetened milk is more consumed than yogurt, while again in Hungary and Germany, sweetened milk is the category of dairy product with the highest consumption with regards T3 in Figure 4.

Descriptive characteristics of the sample are shown in Table 1 and Table 2 in terms of distribution of the samples per surveys for age and sex categories, BMI z-scores, total energy consumption, total dairy intake, and DQI. Besides, in Table 2 and Table 3, the consumption of dairy products at main meals and snack occasions by sex and age categories and by survey, respectively, are shown.

The percentage of energy contribution of dairy products (obtained throughout the 24-HDR) during snack occasions decreased over time (decreased 3% from T0 to T1 and from T1 to T3; from T0 to T3, a 6% energy contribution of dairy products decreased; data not shown).

Boys showed higher energy intakes and total dairy intake at any main meal compared to girls, while they had lower scores of DQI in any survey or age category (Table 2). In Table 3, dairy intakes for total and each snack meal occasions are described for the three survey periods. In general, in T0, the consumption of dairy products at snack occasions was higher than in T1 and even higher than in T3, both in terms of absolute values in grams per day (Table 3) and in terms of relative energy contribution at snack occasions (data not shown). Consumption of milk was higher than yogurt at any snack moment of the day in any survey and in any age or sex category. Besides, the mid-afternoon snack moment was the one in which the children or adolescents consumed higher amounts of dairy products.

In Table 4, children who consumed dairy products at snack occasions had, in general, higher adjusted means of DQI scores apart from those consuming sweetened milk or milky desserts. Specifically, the consumption of the sum of milk and yogurt (*p* = 0.04) and cheese (*p* < 0.001) determined higher DQI scores in T0; yogurt (*p* < 0.001), the sum of milk and yogurt (*p* < 0.001), and cheese (*p* < 0.001) in T1, and only cheese in T3 (*p* = 0.052). In contrary, those consuming sweetened milk in T0 had significantly lower DQI scores (*p* = 0.03).

When analyzing longitudinally the consumption of dairy products at snack occasions and their tracking for each survey (Table 5), results pointed to the same direction. The adjusted tracking of the DQI mean values by groups of dairy consumers and non-consumers throughout T0 and T1 in all snack occasions showed that those who consumed either milk (*p* = 0.02), yogurt (*p* < 0.001), or cheese (*p* < 0.001) determined significantly higher scores of DQI. The same result was observed between the tracking of T1 and T3 for cheese (*p* = 0.03) and for the combined category of milky desserts and sweetened milk with only borderline significance (*p* = 0.054).

## 4. Discussion

The main outcome of the present study showed that children who consumed dairy products at snack meal occasions had a higher diet quality than their non-consumers counterparts, both in the cross-sectional and the prospective analysis. To the best of our knowledge, this is the first study showing such association in such a large sample of children in a prospective cohort. These results seem relevant due to their influence on growth and development in children and adolescents.

There are several definitions of eating occasions in between meals. There is evidence that eating three main meals and two or three snacks per day (nibbling, frequent small intakes) is considered better for health than eating fewer but larger meals (gorging, infrequent large meals) [3,28]. Therefore, a standard definition is required to facilitate comparisons and further research. At the same time, these differences could influence the direction and the magnitude of the associations with diet quality and other health outcomes [29]. Eating occasions in between meals (snacking) can help to control hunger, but snacking should not replace proper meals [28]. Indeed, the snacking habit is commonly associated with poor health outcomes and dietary patterns [30], mainly when the snacks are high-energy dense foods, which might contribute as a factor in the development of childhood overweight and obesity. Although eating at least four meals per day is recommended for children older than 2 years of age [3], some studies have investigated the association between snacking and body mass index and have shown mixed results [31]. The reason could be that their food composition is often poorly described [4]. In this perspective, our study brings some elements showing that dairy products consumption during snack meal occasions might positively affect the overall diet quality.

Our results showed that, in European children, milk and yogurt are generally the dairy most consumed at snack occasions apart from Germany and Hungary, where sweetened milk and milky desserts are mainly consumed, in agreement with previous results [32]. Assessment of children’s diet is of considerable interest because food habits and behavior acquired during childhood and adolescence are used to track until adulthood [33]. In our study, the percentage of energy corresponding to dairy products in relation to the total energy intake consumed at snack occasions decreased over time, as did the absolute values of dairy consumption, except for younger boys between T0 and T1, which may be related to a decrease in the parental influence on children’s dietary patterns [34].

Previous studies in adults suggested that the beneficial effect of dairy products may be dependent on the type of dairy products consumed [35,36,37]. Therefore, given the low dairy products consumption along the day observed in the studied children and adolescents, snack time could be a good opportunity to increase daily intakes of milk and yogurt, specifically among girls who have shown lower consumption of dairy products both in main meals and at snack meal occasions.

Tracking of milk and/or yogurt and/or cheese consumption is also associated with high DQI scores. Other authors have also shown that dairy consumption is linked to healthy dietary patterns and lifestyle factors in adults [38]. This is confirmed in our study.

Children who consumed milk and yogurt had a positive relationship with a combination of high physical activity and low sedentary behaviors over the years [39]. The most likely reason is the clustering of different lifestyle behaviors, such as physical activity, smoking, or dietary behavior [40]. Moreover, the consumption of yogurt and other dairy products has been shown to be associated with a high diet quality, a decreased risk of weight gain and obesity, as well as decreased risk of cardiovascular diseases [41], and it may contribute to a healthier insulin profile in adults [42].

Some studies have suggested that dairy intake and its contribution to calcium intake could have a protective effect on the development of type 2 diabetes, as an inverse association with the incidence of insulin resistance, independently of age and family history of diabetes, was shown [43,44]. A recent review suggests a neutral effect of dairy intake on adiposity during early and middle childhood and a modestly protective effect in adolescence [45]. Dairy intake also contributes to dietary fat consumption; however, there is recent evidence showing the positive health effects of dairy fat [41,46]. There is only one recent cross-sectional study showing that a frequent consumption of yogurt was associated with a better healthy eating index in US children [47]. However, information on the association between dairy consumption and diet quality in children is scarce; for this reason, we are providing further evidence showing that those children consuming higher amounts of dairy products (specifically yogurt and milk) had also higher values of diet quality index and that this association is maintained after two years.

During the last years, an obesogenic environment has been developed [48], leading to an increase of unhealthy eating habits [49], among other lifestyle behaviors components. For instance, a recent review also found an association between watching TV during meals and a high consumption of energy-dense high-fat, high-sugar foods such as savory snacks in children [50]. Therefore, our results show an agreement with available guidelines suggesting a benefit in the replacement of energy-dense foods by dairy products at snack time in terms of diet quality [28].

The present study is subject to a number of limitations. For instance, the fact that the number of participants on each survey decreased progressively may affect our results as, normally, those who are healthier and with higher social position are less likely to drop out. However, in our cohorts, drop-outs showed to slightly distort the distribution of children’s BMI for those above the 99% percentile, affecting marginally exposure-outcome associations [14]. Besides, food consumption was obtained using only a single 24-HDR. Although the use of multiple 24-HDRs may have provided more reliable data, this method is still considered as a valid tool for the estimation of population means, as random errors cancel out on a group level [51]. The diet quality index was obtained from the combination of FFQ and 24-HDR, as it was previously done by other authors [20,21]. In addition, the DQI, based on the FFQ, was previously shown to be a good approach when compared with a 3 days record in Flemish pre-schoolers [21]. Moreover, the sample size was reduced for the prospective analysis, although a considerable sample size was achieved. Regarding the moment of consumption and specifically snacking, our study included any food or beverage intake outside of the usual main meals (breakfast, lunch, and dinner) without the possibility to discriminate structured snacks from nibbling behavior.

The standardized dietary assessment and linkage to country-specific food composition tables using validated 24-HDR software as well as the cooperation with schools, which enabled the assessment of school meal information, are the main strengths of this study. The assessment period covered almost an entire year, taking into account seasonal variations in the diet. Moreover, the large sample size comprises data from eight European countries; the strictly standardized data assessment, documentation, and data cleaning processing guarantee the highest possible data quality.

In conclusion, higher consumption of milk, yogurt, and cheese at snack meal occasions throughout childhood is related to a higher diet quality. Inclusion of dairy products outside the main meals could be a good strategy to decrease the consumption of energy-dense foods in children and adolescents.

## Figures and Tables

**Figure 1 nutrients-12-00642-f001:**
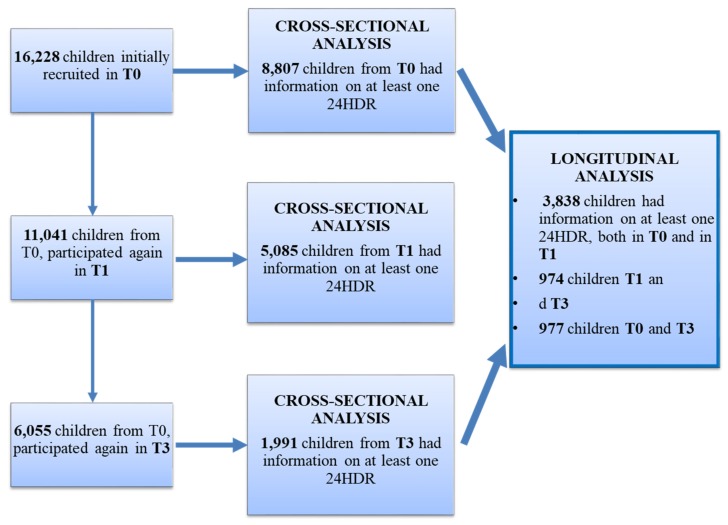
Flow chart of the sampling.

**Figure 2 nutrients-12-00642-f002:**
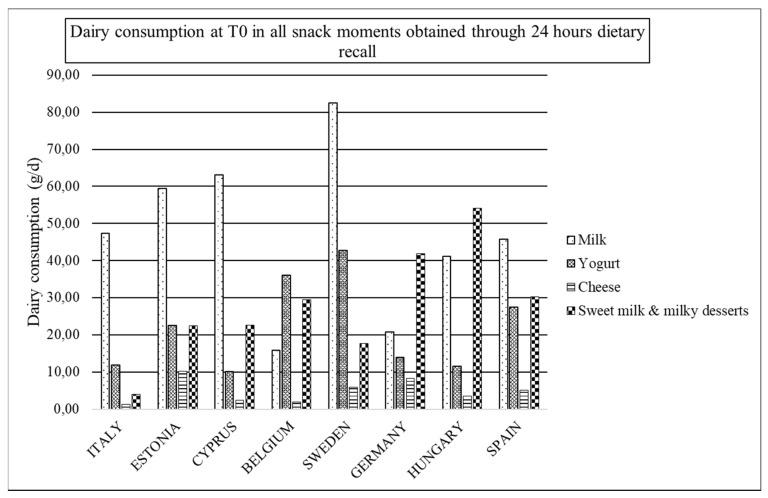
Dairy consumption at T0 in all snack moments obtained through 24 HDR.

**Figure 3 nutrients-12-00642-f003:**
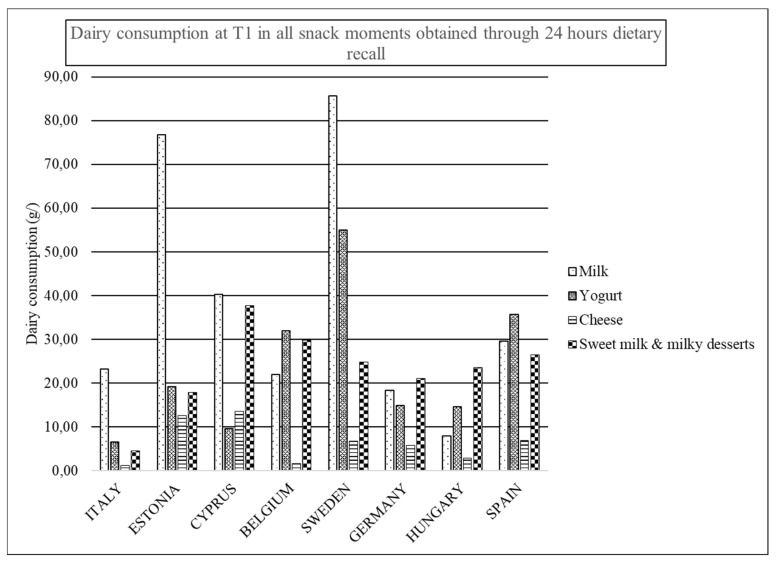
Dairy consumption at T1 in all snack moments obtained through 24 HDR.

**Figure 4 nutrients-12-00642-f004:**
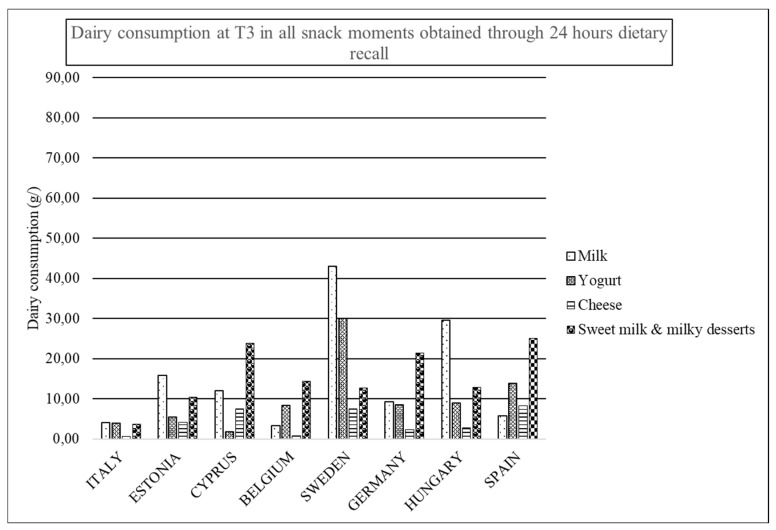
Dairy consumption at T3 in all snack moments obtained through 24 HDR.

**Table 1 nutrients-12-00642-t001:** Descriptive characteristics of the children participating in the different surveys T0, T1, and T3.

	T0				T1				T3			
	Boys		Girls		Boys		Girls		Boys		Girls	
	2–6 y	6–10 y	2–6 y	6–10 y	2–6 y	6–12 y	2–6 y	6–12 y	<10 y	≥10 y	<10 y	≥10 y
**N**	1961	2504	1814	2528	501	2076	445	2063	450	575	418	548
**BMI z-score ***	0.18 (1.18)	0.54 (1.25)	0.21 (1.11)	0.54 (1.17)	0.24 (1.20)	0.61 (1.22)	0.27 (1.19)	0.60 (1.17)	0.50 (1.24)	0.57 (1.28)	0.29 (1.21)	0.34 (1.21)
**Education of the Families (ISCED 2011) T0**												
High education	888	1064	832	1122	253	987	249	991	241	310	260	278
Medium education	905	1220	847	1204	214	931	175	927	188	245	144	241
Low education	168	220	135	202	34	158	21	145	21	20	14	29
**Country**												
Italy	483	558	388	553	129	657	101	612	64	112	50	107
Estonia	131	215	138	254	148	351	131	390	109	95	85	84
Cyprus	168	263	163	273	6	149	9	159	11	30	11	35
Belgium	116	83	105	76	34	166	35	148	28	34	24	38
Sweden	296	326	272	320	70	207	60	204	57	87	74	67
Germany	372	528	369	513	31	206	38	203	38	96	48	66
Hungary	224	392	247	395	17	90	18	87	29	25	20	30
Spain	171	139	132	144	66	250	53	260	45	45	46	40
**Region**												
Control	918	1146	821	1201	250	1088	226	1065	224	241	186	231
Intervention	1043	1358	993	1327	251	988	219	998	226	334	232	317

* This row corresponds to mean and (standard deviations), while the rest of the rows are number of participants (N).

**Table 2 nutrients-12-00642-t002:** Description of dietary information (diet quality, energy, and dairy intakes) at main meals for boys and girls in the different age categories for each survey period.

	T0 (2007–2008)				T1 (2009–2010)				T3 (2013–2014)			
	Boys		Girls		Boys		Girls		Boys		Girls	
	2–6 y	6–10 y	2–6 y	6–10 y	2–6 y	6–12 y	2–6 y	6–12 y	<10 y	≥10 y	<10 y	≥10 y
Diet Quality Index (DQI)	79.97 (16.03)	80.7 (18.04)	80.62 (16.12)	83.22 (17.36)	85.69 (19.09)	85.22 (18.98)	87.75 (18.02)	87.90 (19.32)	99.76 (16.82)	97.86 (18.43)	101.26 (18.56)	100.85 (18.54)
Total energy intake (kcal/day)	1460.25 (509.69)	1675.2 (568.23)	1374.55 (502.57)	1529.91 (512.75)	1634.04 (512.72)	1774.23 (571.07)	1503.72 (516.58)	1619.30 (546.67)	1648.12 (688.47)	1611.99 (650.67)	1485.81 (624.40)	1482.05 (570.48)
Total Dairy intake (g/day)	334.98 (241.84)	318.52 (259.12)	303.78 (232.57)	270.34 (227.84)	368.29 (240.99)	322.90 (258.65)	326.51 (235.56)	282.67 (217.55)	254.35 (234.82)	233.52 (211.30)	239.90 (218.76)	209.64 (218.83)
**Breakfast**												
Milk (g/day)	91.65 (108.40)	81.99 (102.58)	81.50 (109.20)	68.42 (95.64)	93.01 (105.67)	101.01 (109.21)	83.77 (101.28)	85.54 (98.19)	82.39 (105.70)	90.42 (109.42)	74.75 (97.52)	60.73 (86.92)
Yogurt (g/day)	15.88 (54.35)	13.95 (53.02)	13.38 (47.33)	12.01 (47.72)	17.09 (54.97)	14.52 (52.88)	14.71 (46.42)	15.41 (53.01)	13.82 (50.98)	12.81 (51.47)	14.31 (47,47)	12.84 (45.03)
Milk + yogurt (g/day)	107.52 (114.57)	95.93 (109.33)	94.87 (113.38)	80.43 (101.39)	110.10 (112.07)	115.53 (112.43)	98.48 (101.12)	100.95 (102.19)	96.21 (111.50)	103.23 (113.29)	89.06 (104.68)	73.57 (92.78)
Cheese (g/day)	2.81 (14.91)	2.10 (10.87)	2.75 (16.77)	2.01 (10.72)	2.42 (17.89)	2.20 (13.56)	2.22 (13.23)	2.45 (12.74)	2.93 (18.51)	2.91 (11.78)	4.33 (15.82)	2.92 (14.88)
Milky desserts + sweetened milk (g/day)	18.43 (59.55)	18.85 (60.80)	15.0 (55.25)	18.0 (57.38)	9.91 (39.58)	13.27 (50.81)	14.38 (51.88)	14.70 (52.09)	19.11 (62.77)	16.27 (61.13)	17.86 (59.63)	19.53 (60.09)
**Lunch**												
Milk (g/day)	17.94 (54.56)	21.06 (66.53)	16.55 (49.79)	15.57 (56.80)	28.76 (68.95)	20.57 (63.58)	23.03 (63.68)	16.52 (54.93)	19.89 (59.72)	17.22 (60.40)	20.45 (62.79)	13.60 (55.78)
Yogurt (g/day)	7.93 (34.60)	6.80 (32.38)	5.31 (27.95)	6.53 (31.57)	6.12 (30.47)	9.29 (37.73)	7.48 (36.98)	7.89 (33.81)	6.08 (31.51)	6.95 (33.40)	4.95 (26.53)	5.20 (28.56)
Milk + yogurt (g/day)	25.86 (63.77)	27.85 (72.77)	21.86 (56.56)	22.10 (63.71)	34.88 (73.83)	29.86 (72.11)	30.51 (71.31)	24.41 (62.64)	25.97 (67.12)	24.17 (68.12)	25.40 (68.52)	18.80 (61.52)
Cheese (g/day)	4.76 (26.32)	5.14 (21.89)	5.57 (22.08)	6.05 (26.21)	8.93 (27.65)	6.19 (24.24)	7.01 (25.01)	6.0 (23.03)	5.15 (22.16)	3.14 (16.42)	3.29 (16.71)	4.06 (22.16)
Milky desserts + sweetened milk (g/day)	4.76 (26.32)	6.63 (37.93)	4.63 (32.40)	6.04 (29.57)	9.3343 (40.37)	8.91 (39.96)	8.87 (36.58)	8.92 (38.27)	19.11 (62.77)	16.27 (61.13)	17.86 (59.63)	19.53 (60.09)
**Dinner**												
Milk (g/day)	31.82 (77.85)	37.53 (88.72)	29.60 (71.51)	27.85 (74.04)	46.62 (86.76)	33.59 (81.77)	39.13 (82.87)	28.67 (74.64)	29.21 (83.20)	20.52 (63.42)	24.16 (66.39)	23.57 (72.32)
Yogurt (g/day)	11.16 (43.63)	8.33 (37.91)	8.75 (37.16)	6.57 (31.09)	9.56 (39.47)	8.57 (37.95)	9.94 (35.80)	9.60 (38.29)	7.08 (37.16)	7.24 (41.46)	8.06 (37.65)	7.24 (35.13)
Milk + yogurt (g/day)	42.98 (88.38)	45.86 (94.60)	38.35 (78.99)	34.43 (78.51)	56.18 (93.29)	42.16 (87.77)	49.08 (88.711)	38.27 (81.77)	36.29 (89.57)	27.76 (74.37)	32.22 (74.44)	30.80 (80.52)
Cheese (g/day)	5.57 (20.68)	6.35 (23.29)	6.22 (23.95)	6.07 (24.40)	7.39(25.55)	7.80 (27.85)	7.93 (30.45)	6.91 (24.30)	5.54 (19.36)	5.04 (20.79)	5.47 (21.27)	5.06 (21.46)
Milky desserts + sweetened milk (g/day)	10.99 (43.65)	14.01 (52.42)	9.83 (42.30)	9.29 (41.32)	8.27 (34.45)	7.52 (34.48)	6.20 (28.03)	6.99 (31.20)	19.11 (62.77)	16.27 (61.13)	17.86 (59.63)	19.53 (60.09)

All the data in cells correspond to means and (standard deviations).

**Table 3 nutrients-12-00642-t003:** Description of the dietary information at snack meal occasions for boys and girls in the different age categories per survey (in g/day, SD).

	T0 (2007–2008)				T1 (2009–2010)				T3 (2013–2014)			
	Boys		Girls		Boys		Girls		Boys		Girls	
	2–6 y	6–10 y	2–6 y	6–10 y	2–6 y	6–12 y	2–6 y	6–12 y	<10 y	≥10 y	<10 y	≥10 y
**All Snacks**												
Milk	56.19 (107.24)	44.21 (96.46)	53.25 (98.03)	38.27 (85.70)	65.18 (105.80)	41.95 (104.65)	55.3 (100.27)	31.86 (82.24)	20.08 (67.87)	15.30 (60.70)	13.90 (49.47)	12.50 (50.40)
Yogurt	22.82 (64.18)	18.60 (64.27)	20.49 (58.75)	16.25 (53.78)	31.75 (71.75)	21.57 (65.00)	24.79 (61.20)	17.17 (53.32)	11.15 (43.62)	9.10 (43.11)	12.73 (47.15)	9.35 (46.02)
Milk + yogurt	79.0 (123.33)	62.81 (117.68)	73.74 (110.84)	54.52 (101.14)	96.93 (120.76)	63.52 (121.63)	80.09 (114.39)	49.03 (96.46)	31.23 (81.24)	24.39 (73.17)	26.63 (68.95)	21.85 (71.41)
Cheese	4.66 (17.82)	5.19 (19.14)	4.89 (17.53)	4.54 (14.52)	6.11 (24.35)	6.19 (20.82)	6.17 (24.38)	6.07 (20.00)	3.52 (20.47)	3.30 (21.74)	4.39 (28.20)	3.27 (17.63)
Milky desserts + sweetened milk	26.71 (76.55)	27.81 (78.17)	26.06 (73.29)	26.77 (79.95)	17.84 (54.88)	19.74 (64.80)	15.57 (53.75)	17.96 (61.27)	12.18 (50.76)	10.16 (51.12)	14.76 (56.62)	14.70 (54.13)
**Mid-Morning Snack**												
Milk	5.62 (37.97)	7.06 (37.31)	6.74 (31.62)	8.92 (40.37)	15.52 (57.68)	5.22 (31.39)	12.02 (46.36)	3.38 (25.80)	2.65 (25.46)	1.85 (20.79)	1.84 (17.11)	2.53 (25.68)
Yogurt	4.97 (28.56)	3.98 (36.07)	4.28 (24.83)	2.98 (20.92)	5.24 (28.92)	4.63 (27.20)	5.45 (28.57)	3.56 (22.38)	2.61 (20.23)	0.96 (11.86)	1.68 (13.42)	0.55 (7.39)
Milk + yogurt	10.59 (47.47)	11.04 (52.94)	11.01 (39.82)	11.90 (44.88)	20.76 (64.58)	9.85 (41.12)	17.47 (53.23)	6.94 (33.80)	5.26 (32.30)	2.80 (23.86)	3.52 (21.60)	3.078 (26.67)
**Mid-Afternoon Snack**												
Milk	20.0 (59.83)	18.90 (64.26)	19.04 (56.46)	14.66 (53.59)	27.18 (66.59)	20.03 (66.15)	29.70 (70.63)	18.34 (61.91)	10.21 (46.01)	8.62 (46.96)	8.97 (41.74)	6.03 (33.35)
Yogurt	12.74 (45.01)	8.90 (40.01)	12.04 (43.94)	8.43 (37.76)	15.45 (49.04)	10.24 (42.81)	14.25 (46.07)	9.23 (38.52)	5.54 (30.16)	3.73 (26.79)	5.35 (35.12)	5.622 (32.71)
Milk + yogurt	32.74 (74.24)	27.80 (76.23)	31.08 (70.06)	23.10 (65.54)	42.65 (78.59)	30.27 (77.86)	43.96 (83.28)	27.58 (72.05)	15.75 (56.64)	12.35 (54.30)	14.32 (54.10)	11.66 (49.43)
**After-Dinner Snack**												
Milk	30.57 (76.31)	18.25 (58.92)	27.48 (70.75)	14.68 (51.15)	22.46 (67.60)	16.69 (59.16)	13.573 (52.99)	10.14 (45.60)	7.22 (41.18)	4.83 (31.98)	3.09 (22.65)	3.93 (24.63)
Yogurt	5.11 (32.82)	5.72 (34.34)	4.17 (28.86)	4.84 (31.36)	11.05 (46.01)	6.71 (39.05)	5.09 (29.69)	4.38 (29.39)	3.0 (25.17)	4.41 (29.35)	5.70 (30.13)	3.18 (28.98)
Milk + yogurt	35.67 (81.69)	23.98 (67.58)	31.65 (75.45)	19.53 (59.05)	33.52 (80.93)	23.40 (70.36)	18.66 (60.16)	14.52 (54.14)	10.22 (48.25)	9.24 (42.92)	8.78 (37.62)	7.12 (37.70)

All the data in cells correspond to means and (standard deviations).

**Table 4 nutrients-12-00642-t004:** DQI by non-consumers and consumers of the different dairy products considering all snack meal occasions, cross-sectional analyses in T0 and T1.

		T0		T1		T3	
		Means	(CI 95%) *p*-Values	Means	(CI 95%) *p*-Values	Means	(CI 95%) *p*-Values
All Snacks							
Milk (g/day)	NON CONSUMERS (N) CONSUMERS (N)	85.91 (6640) 87.21 (2160)	(−0.06, 1.71) 0.06	86.17 (4082) 88.22 (1002)	(−0.24, 2.39) 0.11	99.73 (1831) 100.88 (160)	(−2.46, 3.40) 0.75
Yogurt (g/day)	NON CONSUMERS (N) CONSUMERS (N)	86.06 (7754) 87.41 (1042)	(−0.43, 1.92) 0.22	85.92 (4429) 90.96 (654)	(2.66, 5.73) <0.001	99.68 (1867) 101.96 (124)	(−1.68, 4.88) 0.34
Milk + yogurt (g/day)	NON CONSUMERS (N) CONSUMERS (N)	85.74 (5843) 87.18 (2957)	(0.06, 1.68) 0.04	85.47 (3538) 89.09 (1546)	(1.44, 3.71) <0.001	99.60 (1720) 101.27 (271)	(−1.32, 3.34) 0.40
Cheese (g/day)	NON CONSUMERS (N) CONSUMERS (N)	85.34 (7453) 88.35 (1346)	(1.02, 3.13) <0.001	85.92 (4262) 89.95 (821)	(1.39, 4.21) <0.001	99.56 (1845) 103.16 (146)	(−0.03, 6.06) 0.052
Milky desserts (g/day)	NON CONSUMERS (N) CONSUMERS (N)	86.25 (8203) 85.81 (593)	(−2.69, 0.35) 0.13	86.50 (4709) 87.40 (374)	(−1.56, 2.39) 0.68	99.93 (1869) 98.18 (122)	(−5.32, 1.28) 0.23
Sweetened milk (g/day)	NON CONSUMERS (N) CONSUMERS (N)	86.32 (7405) 85.71 (1392)	(−2.21, −0.12) 0.03	86.38 (4442) 87.90 (641)	(−0.87, 2.25) 0.39	99.75 (1930) 102.33 (61)	(−2.95, 6.25) 0.48

95% confidence intervals and *p*-values correspond to linear regression analyses adjusted by BMI z-score (age and sex adjusted), energy intake, higher education of the parents, and type of region only for T1 and T3 (control or intervention).

**Table 5 nutrients-12-00642-t005:** Diet Quality Index by groups according to tracking of dairy products consumption at snack meal occasions between T0 and T1, T1 and T3, T0 and T3.

		T0–T1		T1–T3		T0–T3	
		Mean DQI (*n*)	(CI 95%) *p*-Values	Mean DQI (*n*)	(CI 95%) *p*-Values	Mean DQI (*n*)	(CI 95%) *p*-Values
**All Snacks**							
Milk	UNHEALTHY TRACKING HEALTHY TRACKING	85.46 (3039) 88.11 (799)	(0.26, 3.20) 0.02	98.85 (908) 101.27 (66)	(−2.35, 6.88) 0.34	98.74 (914) 100.71 (77)	(−3.12, 5.20) 0.62
Yogurt	UNHEALTHY TRACKING HEALTHY TRACKING	85.49 (3388) 89.91 (450)	(1.70, 5.36) <0.001	98.86 (921) 101.73 (53)	(−3.14, 7.05) 0.45	98.79 (928) 100.47 (63)	(−4.06, 5.0) 0.84
Milk + yogurt	UNHEALTHY TRACKING HEALTHY TRACKING	84.90 (2675) 88.56 (1163)	(1.36, 3.96) <0.001	98.65 (860) 101.75 (114)	(−0.95, 6.26) 0.15	98.59 (859) 100.85 (132)	(−2.09, 4.48) 0.48
Cheese	UNHEALTHY TRACKING HEALTHY TRACKING	85.14 (3235) 89.59 (603)	(1.74, 5.0) <0.001	98.63 (913) 104.85 (61)	(0.43, 9.95) 0.03	-	-
Milky desserts + sweetened milk*	HEALTHY TRACKING UNHEALTHY TRACKING	86.34 (450) 85.97 (3388)	(−1.36, 2.31) 0.61	96.81 (120) 99.33 (854)	(−0.07, 6.96) 0.054	98.11 (85) 98.97 (906)	(−2.61, 5.27) 0.51

95% confidence intervals and *p*-values correspond to linear regression analyses adjusted by BMI z-score (age and sex adjusted), energy intake, higher education of the parents in the final survey (T1 or T3 accordingly), and type of region only for T1 and T3 (control or intervention). Healthy tracking: consumers in both surveys, non-consumers in the first survey but consumers in the final survey (apart from the case for milky desserts and sweetened milk, for which it was the opposite). Unhealthy tracking: non-consumers in both surveys, consumers in the first survey but non-consumers in the final survey (apart from the case for milky desserts and sweetened milk, for which it was the opposite). There are no data for cheese in T0–T3 because the unhealthy tracking group was empty.

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
