# Peer review of "Dairy Consumption at Snack Meal Occasions and the Overall Quality of Diet during Childhood. Prospective and Cross-Sectional Analyses from the IDEFICS/I.Family Cohort"

_nutrients, 2020, doi:10.3390/nu12030642_

Round 1

Reviewer 1 Report

This manuscript is well written, and the topic is interesting and highly relevant to the human nutrition and health. It investigates the association of overall diet quality and consumption of dairy products at snack time through childhood. Methodology has been well described with sufficient information. Few suggestions to further improve the quality of this manuscript;

It would be better to include a brief description about popular dairy products which consume as snacks (yogurt, cheese, sweet milk and other minor products) in the introduction.

Pay attention on the minor typos and grammatical errors and rectify those please.

Overall, high quality manuscript and no objections in publishing this manuscript in high impact journal-nutrition.

Author Response

Enclosed you will find a revision of our manuscript “Dairy consumption at snack meal occasions and the overall quality of diet during childhood. 1 Prospective and cross-sectional analyses from the IDEFICS/ I.Family cohort”. We would like to thank the reviewers for their thoughtful and constructive comments. We have considered all of the suggestions and have incorporated them into the revised manuscript (highlighted in yellow). We believe our manuscript is stronger as a result of these modifications. An itemized point-by-point response to the reviewers’ comments is presented below.

This manuscript contains material that is original and not previously published in text or on the Internet, nor is it being considered elsewhere until a decision is made as to its acceptability by the Nutrients Editorial Review Board.

Reviewer #1 (Comments to the Author)

This manuscript is well written, and the topic is interesting and highly relevant to the human nutrition and health. It investigates the association of overall diet quality and consumption of dairy products at snack time through childhood. Methodology has been well described with sufficient information. Few suggestions to further improve the quality of this manuscript.

Overall, high quality manuscript and no objections in publishing this manuscript in high impact journal-nutrition.

Comment 1

It would be better to include a brief description about popular dairy products which consume as snacks (yogurt, cheese, sweet milk and other minor products) in the introduction.

Answer 1

Comments appreciated. A brief description of dairy products, yogurt, cheese, milky desserts and sweetened milk were included in the introduction sections.

 “The main dairy products consumed are 1) yogurt which is defined as a food in the form of a thick, slightly sour liquid that is made by adding bacteria to milk and 2) cheese. Other dairy product consumed were included in a milky dessert which are a pooled group of foods contains a lot of milk, and sweetened milk which have an added sugar portion inside. Dairy products are good sources of calcium, which helps maintain bone mineral content and could reduce the risk of fractures later in life [5]. ”

  1. van den Heuvel EGHM, Steijns JMJM. Dairy products and bone health: how strong is the scientific evidence? Nutr Res Rev. 2018 Dec;31(2):164-178.

Comment 2:

Pay attention on the minor typos and grammatical errors and rectify those please.

  • Answer 2

Thank you for the comments. Minor typos and grammatical errors were rectified across the manuscript. Moreover, taking into consideration the reviewer’ s comments we include one reference and update the reference list.

Reviewer 2 Report

General comments:

Overall, the manuscript by Iglesia et al. is interesting and well-written. The authors did prospective and cross-sectional analyses of children who consumed dairy products at snack meal occasions from the IDEFICS/I.Family cohort. It is a novel study showing the potential benefits (e.g., a higher diet quality) of higher consumption of milk, yogurt and cheese at snack meal occasions through out childhood. Due to the complexity of experimental design, several limitations were discussed by authors. Particularly, the sample size has be decreased from T0 to T3 and it is unclear that how many children were consistently involved in the study? The authors need to add the information about how many kids were included for one time, two times, and three times. Another limitation is that only one single 24-HDR data was considered for each participant. Many other factors from the day before or even several days before could affect the results. Besides these concerns, a few minor points need to be addressed. After a minor revision, the manuscript should be good for publication at Nutrients.

Specific comments are listed below:

Minor points:

Line (L) 36: Please explain why the number of participants decreased significantly from T0-T3?

L76: Diet quality indices (DOIs) need to be better defined.

L118: Figure 1: There is a typo in the box ‘Longitudinal Analysis’-  974 children in T1 and T3’. How many children were included in T0, T1, and T3?

L149: Define ‘Cross-sectional analyses’. I would suggest using calories per day instead of in grams per day for intake.

L151: It is unclear how many times for each child consume milk per day. Some of them might just consumed one time per day, others might consumed two or three times per day.

L160 -162: Among these 43 food items in T0/T1 and 60 food items in T3, how many of them are the same?

L229: How did you define the healthy and unhealthy tracking groups?

L242: Figure 2, did you do statistical comparisons? If yes, please mark the significances and P values in the figure.

L254-255 and L 261: Why the data are not shown? The decreased percent (e.g., 19% in T0?) was compared to what?

L267: Table 2: Did you do statistics? Which ones are significant?

L288: Table 4: Typo for Milk + yogurt (<0.00?)

Author Response

 Enclosed you will find a revision of our manuscript “Dairy consumption at snack meal occasions and the overall quality of diet during childhood. 1 Prospective and cross-sectional analyses from the IDEFICS/ I.Family cohort”. We would like to thank the reviewers for their thoughtful and constructive comments. We have considered all of the suggestions and have incorporated them into the revised manuscript (highlighted in yellow). We believe our manuscript is stronger as a result of these modifications. An itemized point-by-point response to the reviewers’ comments is presented below.

This manuscript contains material that is original and not previously published in text or on the Internet, nor is it being considered elsewhere until a decision is made as to its acceptability by the Nutrients Editorial Review Board.

Reviewer #2 (Comments to the Author)

  • Comment 3
  • Overall, the manuscript by Iglesia et al. is interesting and well-written. The authors did prospective and cross-sectional analyses of children who consumed dairy products at snack meal occasions from the IDEFICS/I.Family cohort. It is a novel study showing the potential benefits (e.g., a higher diet quality) of higher consumption of milk, yogurt and cheese at snack meal occasions throughout childhood. Due to the complexity of experimental design, several limitations were discussed by authors. Particularly, the sample size has be decreased from T0 to T3 and it is unclear that how many children were consistently involved in the study? The authors need to add the information about how many kids were included for one time, two times, and three times.
  • Another limitation is that only one single 24-HDR data was considered for each participant. Many other factors from the day before or even several days before could affect the results. Besides these concerns, a few minor points need to be addressed. After a minor revision, the manuscript should be good for publication at Nutrients.

Answer 3

The authors appreciate the comment from the Reviewer. This has been clarified as suggested across the manuscript, in the abstract, material and method and discussion sections.

Comment 4: Minor points:

Line (L) 36: Please explain why the number of participants decreased significantly from T0-T3?

Comments appreciated. In order to avoid duplicating information, the number of participants included in T0, T1, and T3 were included in the abstract (L39) and material and method (L103-112: baseline sample, attrition L129-131 and L131-132: sample included in the analysis).

The attrition rate was defined by Langeheine et al [14]. Children’s age and weight status was positively associated with attrition; however, mother’s age, migrant background and educational level were associated with lower attrition.

“Taking into consideration the availability of the variables for this study (age, sex, BMI z-scores, education of the families, region, valid data on the two dietary assessment methods 24HDR and FFQ), total samples in each of the surveys were 8,807 in T0, 5,085 in T1, and 1,991 children in T3 for cross-sectional analyses. In relation to longitudinal analyses, we finally included: 3,838 children for tracking between T0 and T1; 974 children for tracking between T1 and T3; and 977 children with available data to assess the tracking between T0 and T3.

L76: Diet quality indices (DOIs) need to be better defined.

Comments appreciated. DQI definition was completed in the Material and Methods section (L194-198). Moreover, a global assessment of DQI were also included in the manuscript.

These three components of the DQI are presented in percentages. The dietary quality component ranged from -100 to 100%, while dietary diversity and dietary equilibrium ranged from 0 to 100%. To compute the DQI, the mean of these components was calculated. As such, the DQI ranged from 33 to 100%, with higher scores reflecting higher diet compliance. The score was calculated at baseline, T1 and T3 surveys. DQI scores for an individual provide an estimate of diet quality relative to dietary guidelines.

L118: Figure 1: There is a typo in the box ‘Longitudinal Analysis’-  974 children in T1 and T3’. How many children were included in T0, T1, and T3?

Comments appreciated. In order to avoid duplicating information, the number of participants included in T0, T1, and T3 were included in the abstract (L39) and material and method section (L103-112: baseline sample, attrition: L129-131 and L131-132: sample included in the analysis).

L149: Define ‘Cross-sectional analyses’. I would suggest using calories per day instead of in grams per day for intake.

Comments appreciated. Cross-sectional analyses were defined in the manuscript. Reagarding the grams per day, we decided to use the grams of the corresponding dairy groups instead of the calories because we considered it was important for the statistical analyses to have the kcal as a covariate and also, avoid multicollinearity. Besides, dairies, are food groups with a very high variability in kcal. By using the total kcal provided by them as an independent variable in these analyses, we and also the readers may not have had a real idea of differences in the portions of consumption of each of the groups (for instance, the kcal provided to one child who consumes a small portion of “unhealthy” dairies and a big portion of “healthy” dairies may be the same for each group, and the results of the analyses could suggest no associations (as the kcal are the same), while we are getting some using grams).

L151: It is unclear how many times for each child consume milk per day. Some of them might just consumed one time per day, others might consumed two or three times per day.

Comments appreciated. The food consumption focusing on dairy products was described at snack meal occasions (mid-morning, mid-afternoon, after-dinner) to investigate their relationship with the overall diet quality. Moreover, the statistical analysis include the median taking into consideration ages and gender in our sample. Therefore, the food frequency consumption to achieve the aims is measuered as grams insted of frecuency.

L160 -162: Among these 43 food items in T0/T1 and 60 food items in T3, how many of them are the same?

Comments appreciated. The FFQ used in T3, include the food items from T0 and add 17 food items. This information is inlcuded in the material and methos section in the manuscript (L182-187).

the CEHQ included 43 food items which were clustered into 36 according to their nutritional profiles as it has been previously described [18] and also to make them comparable to the food categories of the 24HDR. Similarly, in T3, the FFQ included the food items from T0 and add 17 food items due to older children and adolescents usually eat a wider range of food groups than the youngest. All food items were also re-arranged with the same nutritional profile purpose, in 43 categories.

L229: How did you define the healthy and unhealthy tracking groups?

Comments appreciated. Healthy and unhealthy tracking groups were defined in the statistical analysis section in the manuscript (L260-264).

Healthy tracking: consumers in both surveys, non-consumers in first survey but consumers in final survey (*apart from the case for milky desserts and sweetened milk for which is the opposite) and unhealthy tracking: non-consumers in both surveys, consumers in first survey but non-consumers in final survey (*apart from the case for milky desserts and sweetened milk for which is the opposite).

L242: Figure 2, did you do statistical comparisons? If yes, please mark the significances and P values in the figure.

Comments appreciated. We were adviced by a statistitian that the analysis should be focused on the main hypothesis, which should result in a reduction of statistical tests and included variables. For this reason, we agreed not to include p-values in the descriptive tables or figures. Other wise, we should have been included a multiple testing test and we have such a numerous comparisons in the tables that we wouln´t find any statistical significance.

L254-255 and L 261: Why the data are not shown? The decreased percent (e.g., 19% in T0?) was compared to what?

Comments appreciated. The data were not shown to focus in the aims. However, the information is completed in the resutls section to avoid lack of information (L289-291).

The percentage of energy contribution of dairy products (obtained throughout the 24-HDR) during snack occasions decreased over time (decreased 3% from T0 to T1 and also from T1 to T3, from the T0 to T3 a 6% of energy contribution of dairy products was decreased; data not shown).

L267: Table 2: Did you do statistics? Which ones are significant?

Comments appreciated. We were adviced by a statistitian that the analysis should be focused on the main hypothesis, which should result in a reduction of statistical tests and included variables. For this reason, we agreed not to include p-values in the descriptive tables or figures. Other wise, we should have been included a multiple testing test and we have such a numerous comparisons in the tables that we wouln´t find any statistical significance.

L288: Table 4: Typo for Milk + yogurt (<0.00?)

Answer 7

The authors appreciate the comment from the Reviewer. The p-values were reviewed and completed in the results section and tables.
